# Topology Optimization-Based Localized Bone Microstructure Reconstruction for Image Resolution Enhancement: Accuracy and Efficiency

**DOI:** 10.3390/bioengineering9110644

**Published:** 2022-11-03

**Authors:** Jisun Kim, Jung Jin Kim

**Affiliations:** Department of Mechanical Engineering, Keimyung University, Daegu 42601, Korea

**Keywords:** topology optimization, bone microstructure, resolution enhancement, trabecular alignment, morphometric indices, proximal femur

## Abstract

Topology optimization is currently the only way to provide bone microstructure information by enhancing a 600 μm low-resolution image into a 50 μm high-resolution image. Particularly, the recently proposed localized reconstruction method for the region of interest has received much attention because it has a high possibility to overcome inefficiency such as iterative large-scale problems of the conventional reconstruction. Despite the great potential, the localized method should be thoroughly validated for clinical application. This study aims to quantitatively validate the topology optimization-based localized bone microstructure reconstruction method in terms of accuracy and efficiency by comparing the conventional method. For this purpose, this study re-constructed bone microstructure for three regions of interest in the proximal femur by localized and conventional methods, respectively. In the comparison, the dramatically reduced total progress time by at least 88.2% (20.1 h) as well as computational resources by more than 95.9% (54.0 gigabytes) were found. Moreover, very high reconstruction accuracy in the trabecular alignment (up to 99.6%) and morphometric indices (up to 2.71%) was also found. These results indicated that the localized method could reconstruct bone microstructure, much more effectively preserving the originality of the conventional method.

## 1. Introduction

The bone microstructure is a major determinant of bone strength and density [1]. Considering the bone microstructure is essential for accurate bone health diagnosis. Bone microstructure-based assessment can analyze mechanical bone strength with up to 94% accuracy [2,3]. However, the current clinical field cannot adequately utilize microstructure information regarding bone quality owing to the limited resolution of imaging modalities. Routine computed tomography (CT) provides in vivo images with a pixel resolution of approximately 600 μm [4], which is insufficient for representing trabecular bone microarchitecture with a thickness of 50–100 μm [5,6,7]. Currently, bone diagnosis relies on bone quantity information that can represent bone strength with approximately 60% accuracy [1,8]. Hence, innovative approaches to obtain bone microstructure information are required to address the limitations of current imaging techniques and minimize the risk of misdiagnosis and/or overdiagnosis of bone health.

Image resolution enhancement provides bone microstructure information by enhancing a low-resolution (LR) image into a high-resolution (HR) image. This enables the use of existing medical imaging modalities without increasing the radiation dose. Image resolution enhancement techniques are largely classified into image processing-based, neural network-based, and topology optimization-based methods. The first method, used for decades, is defined as the conventional method. Image processing-based resolution enhancement focuses on obtaining clearer skeletal images via denoising [9], sharpening [10], deblurring [11], and contrast enhancement [12]. It demonstrates very fast and easy characteristics as rule-based approaches; however, they also show lower robustness for enhancement performance based on the various conditions of LR input images. Hence, image processing-based approaches have not yet implemented routine resolution skeletal image reconstruction into a high-resolution bone microstructure image.

With the emergence of deep learning, artificial neural network-based image resolution enhancement techniques have shown remarkable potential. AlexNet [13], VGGNet [14], UNet [15], ResNet [16], and SRCNN [17] are some of the representative approaches. They find and use key features for image resolution enhancement by learning the correlation between LR and HR images. Their superiority has been widely recognized and applied in various fields [18,19,20] including medicine [21,22]. However, artificial neural network-based approaches have not yet been used for bone microstructure reconstruction due to insufficient training data. Note that artificial neural networks inherently depend on the quantity and quality of training data. While LR images, or neural network input, are sufficiently large and easily obtainable in the clinical field, in vivo HR bone microstructure images, or a neural network target (i.e., output), are few and rarely acquired owing to excessive radiation dose [23], except for cadavers [24,25] or animals [26,27].

Alternatively, topology optimization-based bone microstructure reconstruction [28] has recently attracted considerable attention. This approach is based on the physiological principle of bone remodeling metabolism (i.e., Wolff’s law [29]: self-optimizing capabilities), in contrast to the aforementioned approaches. This method reconstructs the bone microstructure at selected sites (e.g., the region and volume of interest) to achieve maximum strength and minimum pixel-wise density difference via topology optimization. To the best of our knowledge, topology optimization is the only method for bone microstructure reconstruction that successfully reconstructs low-resolution clinical images (e.g., CT scan images) into high-resolution bone microstructure images [28]. However, it requires excessive computational resources, such as large-scale and iterative finite element (FE) analysis. This method requires at least 20 h for a 2D image reconstruction, which is exacerbated for 3D image reconstruction. Therefore, topology optimization-based reconstruction requires groundbreaking improvement in computational efficiency for application in the clinical field.

The computational burden of the conventional topology optimization-based method is due to the structural behavior calculation of a global model (i.e., the entire skeletal system), including the region of interest. The calculation, which is a large-scale problem, is performed in every iteration, and its complexity increases as the size of the global model increases and the dimension expands. Thus, if the localized model is constructed by extracting the ROI from the global model, and only the ROI is reconstructed without considering the other skeletal regions, its computational efficiency can be significantly improved. Based on this requirement, a recent study [30] proposed localized bone microstructure reconstruction with physiological local load estimation [31] only for the ROI. However, this localized approach did not deal with the efficiency and accuracy of reconstruction compared to the conventional approach. Therefore, quantitative and thorough validation should be performed so that localized bone microstructure reconstruction based on topology optimization can be utilized reliably in the clinical field.

This study aims to quantitatively validate the topology optimization-based localized bone microstructure reconstruction method in terms of accuracy and efficiency by com-paring the conventional method. Hence, the bone microstructure was first reconstructed for an ROI based on topology optimization using the conventional method with a global model and global loads. Next, the bone microstructure for the ROI was reconstructed based on topology optimization using the localized method with a localized model and estimated local loads. Finally, the reconstructed results were compared for computational efficiency and reconstruction accuracy.

## 2. Materials and Methods

This study validates localized bone microstructure reconstruction based on topology optimization in three steps (Figure 1). Section 2.1 presents the conventional topology optimization-based bone microstructure reconstruction for generating bone microstructures (i.e., control group) for the ROIs. Section 2.2 presents the novel topology optimization-based bone microstructure reconstruction for generating localized bone microstructures (i.e., experimental group) for the same ROIs. Lastly, Section 2.3 describes the quantitative comparison of the results generated by the two methods.

### 2.1. Conventional Topology Optimization-Based Bone Microstructure Reconstruction Using the Global Model

The primary principle of bone microstructure reconstruction using topology optimization is based on the similarity in structural behavior between continuum-level and micro-level models. A previous study reported that the stress and strain distributions of a micro-level containing bone microstructure could be reproduced well using continuum models [32]. Accordingly, the conventional method reconstructs the bone microstructure of the ROI based on topology optimization by considering the structural behavior of the global model under global loads. Note that the global model is an FE model representing the skeletal system, including cortical bone and cancellous bone, captured in LR. Global loads are external loads induced by the muscle force applied to the cortical bone of the global model.

The conventional approach to bone microstructure reconstruction consists of two steps: mesh refinement and topology optimization. First, mesh refinement constructs a resolution-enhanced global model by dividing each element into n × n sub-elements as small as the representable size of the bone microstructure. Note that the element size is smaller than the average thickness of the trabecular bone of 100 μm. The elastic moduli of the sub-elements constituting the ROI and the other region are calculated using Equations (1) and (2), respectively. Equation (1) refers to a solid isotropic material using the penalization (SIMP) method [33] to represent the micro-level material property. Equation (2) is the bone mineral density (BMD)–elastic modulus relationship in the literature to represent the continuum-level material property [32]. The Poisson’s ratio of all the elements was 0.3 [32].
(1)$$E_{i}=\rho_{i}^{\gamma }E_{0}$$
(2)$$\begin{array}{l} E_{i}=0.3044{\left(2\rho_{i}\right)}^{1.49}E_{0}\;\;\;\;\;\;if\;\;\;\rho_{i}\leq 0.84\\ E_{i}=0.1908{\left(2\rho_{i}\right)}^{2.39}E_{0}\;\;\;\;\;\;if\;\;\;\rho_{i}>0.84 \end{array}$$
where $\rho_{i}^{\gamma }$ represents the relative density of the *i*th finite element, $E_{0}$ denotes the reference elastic modulus set to 15 GPa, and γ is the penalization exponent of 3.

Second, topology optimization reconstructs the bone microstructure in the ROI by iteratively updating the bone distribution at the micro level. This update continues until the bone microstructure has maximum mechanical efficiency, while maintaining patient-specific bone distribution information at the continuum level (Wolff’s law [29]). This step achieves maximum mechanical efficiency and patient information preservation by minimizing ROI compliance and density deviation between the reconstructed and input bones. Note that the mechanical efficiency was evaluated via large-scale FE analysis of the global model with every update. The topology optimization can be formulated as follows:(3)$$\mathrm{Minimize}\;\;\;f(\rho )=\sum_{j=1}^{J}c_{j}(\frac{1}{2}u_{j}^{T}Ku_{j}^{})$$
(4)$$\mathrm{Subject}\;\mathrm{to}\;g(\rho )=\frac{1}{N}{\left\|\rho -\rho_{0}^{}\right\|}^{2}\leq \varepsilon$$
where f(ρ) is the compliance, $c_{j}$ is the weight under load condition j, $u_{j}$ is the displacement under load condition j, and K is the stiffness matrix. ρ denotes the density matrix of the ROI set as a design variable, and a total of N variables are set. ρ has a value between 0.01 (i.e., bone marrow) and 1 (i.e., completely filled bone). ρ and $\rho_{0}$ are the relative density matrices of the reconstructed and original models, respectively. ε is a small constant value, set at 0.01, and N is the total number of finite elements in the ROI.

### 2.2. Novel Topology Optimization-Based Localized Bone Microstructure Reconstruction Using the Localized Model

The primary principle of localized bone microstructure reconstruction using topology optimization is identical to that of the conventional method described in Section 2.1. The chief differentiator of the novel method is the localization of the ROI, which contributes to computational burden reduction. Accordingly, the novel approach for localized bone microstructure reconstruction consists of three steps: localization, mesh refinement, and topology optimization.

First, localization constructs a localized FE model for the ROI, which includes simple ROI extraction from the global model and complex estimation of physiological local loads for the ROI. Extraction of the ROI can be implemented anywhere within the global model, including the cortical bone region. The physiological local loads for the ROI were calculated using static condensation in the FE analysis using the method proposed previously [31]. First, the cut boundary displacements of the ROI are calculated via FE analysis using the same global model and global loads as the conventional method. Then, the reaction forces at the cut boundary are calculated by applying the calculated displacement to the boundary of the ROI. Finally, the physiological local loads for the ROI are estimated, as shown in Equation (5). Note that the resultant reaction forces in the cut boundary and local loads in the ROI should be zero via static condensation.
(5)$$\begin{array}{l} \left\{\mathrm{Local}\;\;\mathrm{load}\;\mathrm{for}\;\mathrm{the}\;\mathrm{ROI}\right\}\;=\left\{F_{c}\right\}\\ \;\;\;\;\;\;\;\;\;\;\;\;\;\;\;\;\;\;\;\;\;\;\;\;\;\;\;\left\{{F^{\prime }}_{c}\right\}≃{K^{\prime }}_{cc}\left\{D_{c}\right\}-{K^{\prime }}_{cl}K_{ll}^{-1}{K^{\prime }}_{lc}\left\{D_{c}\right\} \end{array}$$
where, $F_{c}$ and ${F^{\prime }}_{c}$ are the cut boundary forces from the global and localized models, respectively. K is the stiffness matrix, and the subscripts g, c, and l indicate the global model, cut boundary, and local model, respectively. The K and $K^{\prime }$ matrices are derived from the global and localized models, respectively. $D_{c}$ is the displacement of the cut boundary of the ROI and is obtained by the computational analysis of the global model.

Mesh refinement and topology optimization are the same as in the previous methods. Mesh refinement divides the elements of the localized FE model into sub-elements. The material properties of the localized FE model are given by Equation (1). Topology optimization reconstructs the localized FE model by updating the density of each element using Equations (3) and (4). Note that the two steps in the novel method use the localized FE model and estimate local loads with improved computational resources compared to the conventional method.

### 2.3. Numerical Validation Based on Quantitative Comparison Using Proximal Femur

This study used the proximal femur with a resolution of 600 μm as the LR input for both the methods being compared. The proximal femur has been widely used in many studies as a validated model to investigate the internal structure with perimeter control [34], structural behavior in scaffold-implanted bone [4], interactions between solitary waves and the bone [35], and trabecular alteration in aging [36].

This study also set three ROIs for reconstruction: femoral head, neck, and intertrochanteric region. They have characteristic trabecular patterns, as shown in Figure 2. The femoral head [37], femoral neck, and intertrochanteric region [38] are generally considered significant for assessing skeletal diseases. The three ROIs sizes were set for this study: 4.8 × 4.8 mm^2^ (8 × 8 pixels), 9.6 × 9.6 mm^2^ (16 × 16 pixels), and 14.4 × 14.4 mm^2^ (24 × 24 pixels).

Further, this study reconstructed bone microstructures of 50 μm resolution. It was demonstrated that finite element analysis for human cancellous bone tissue reaches a convergence with a very slight difference of 1.65% at a smaller resolution than 156 μm [39]. Based on the facts, this study carefully determined the element size to be 50 μm, which is much smaller than the size validated in the previous studies.

The bone microstructures were reconstructed under the three daily activity loads (i.e., one-legged stance, abduction, and adduction) described previously [40,41] as the global loads. The three loads have hip contact forces and abductor muscle forces of different magnitudes and directions as shown in Figure 3. For one-legged stance, abduction, and adduction, 2317 N, 1158 N, and 1548 N of hip contact forces were applied toward the center of the femoral head with 24°, −15°, and 56° angles against the vertical axis. Moreover, 703 N, 351 N, and 468 N of abductor muscle forces were applied toward the center of the greater trochanter with 28°, −8°, and 35° angles against the vertical axis. The hip contact forces and abductor muscle forces were applied to the femoral head and greater trochanter sur-face as distributed form to avoid stress concentration problems. The normalized weighting factors (c*_j_* in Equation (3)) for the one-legged stance (6000 cycles per day), ab-duction (2000 cycles per day), and adduction (2000 cycles per day) were set at 0.6, 0.2, and 0.2, respectively.

Finally, computational efficiency and reconstruction accuracy were analyzed for validation. The former was evaluated by measuring the required computational resources and total progress time for the reconstruction. The latter was evaluated using trabecular alignment and morphometric indices [7]. The alignment indicates how well the localized bone microstructure reconstruction expresses the characteristic pattern of a trabecular bone. Morphometric indices quantify how accurately the method can reconstruct the bone microstructure. This study used bone volume fraction (BV/TV), trabecular thickness (Tb.Th), trabecular separation (Tb.Sp), and trabecular number (Tb.N) as the indices.

All calculations in this study were performed on a personal computer (Intel Core™ i9-10900K, 3.70 GHz, 128 GB RAM, Santa Clara, CA, USA). ANSYS 2021 R2 was used for all FE analyses, and the preconditioned conjugate gradient (PCG) method [42] was used as the FE equation solver. This study also used the method of moving asymptotes (MMA) [43] as an optimizer.

## 3. Results

Table 1 shows the computational resources required and the total progress time of bone microstructure reconstruction. The localized method required significantly fewer computational resources compared to the conventional method. The conventional method required very excessive memory of 56.2 GB which is the sum of the used memory for the three loading conditions. Particularly, the amount of required huge memory was the same in all the bone microstructure reconstruction processes. This is the representative problem of the conventional method that used the same global FE model regardless of the size and location of the ROIs. Conversely, the localized method required very small memories for the bone microstructure reconstruction of the same ROIs as follows: 0.3 GB (4.8 × 4.8 mm^2^), 1.0 GB (9.6 × 9.6 mm^2^), and 2.3 GB (14.4 × 14.4 mm^2^). These requirements are due to the localization of the ROIs (Figure 2), which reduced the total number of elements required from 2,149,488 (94.2 × 104.4 mm^2^) to 9216 (4.8 × 4.8 mm^2^), 36,864 (9.6 × 9.6 mm^2^), and 82,944 (14.4 × 14.4 mm^2^). Moreover, the localized method showed that the required memory increased almost squarely as the ROI size increased due to the two-dimensional FE model, unlike the conventional method. Note that the absolute required memory significantly reduced by at least 95.9% compared to the conventional method, although it increased according to the ROI size.

Comparing the total progress time, the localized method showed dramatic improvement in computational efficiency by at least 88.2%. The localized method required a maximum of 2.7 h while the conventional method required a minimum of 13.2 h. Along with reduced memory, the progress time per iteration for reconstruction greatly reduced from 4.08 min (94.2 × 104.4 mm^2^) to 0.18 min (4.8 × 4.8 mm^2^), 0.18 min (9.6 × 9.6 mm^2^), and 0.24 min (14.4 × 14.4 mm^2^). Interestingly, the localized method converged more than the conventional method via iterations, except for the intertrochanter for the ROI of size 14.4 × 14.4 mm^2^, although the optimization formulation and ROI are the same for the two methods. For example, the localized and the conventional methods required 607 and 351 iterations for the reconstruction of the same femoral head of 14.4 × 14.4 mm^2^, respectively. It should be emphasized that the localized method needs only about 2 h despite the increase in iterations.

Figure 4 shows that the two methods reconstructed nearly identical trabecular microarchitectures. Most of the primary trabecular patterns are very similar with a minimal alignment angle difference of less than 2.1°. The maximum deviations in alignment angle according to the location of ROIs are 0.7° (femoral head of 9.6 × 9.6 mm^2^), 2.1° (femoral neck of 14.4 × 14.4 mm^2^), and 1.9° (intertrochanter of 4.8 × 4.8 mm^2^), respectively. The error rates of these angle differences are 2.0%, 2.1%, and 2.1%, respectively. Further, the reconstruction accuracy of alignment is well maintained regardless of location and size. A small difference is also observed in the detailed pattern of the small branches among the main trabeculae. Although, the overall pattern is almost the same. Unlike the conventional method, the localized method tends to reconstruct bone microstructures at each node of the elements on the cut boundary of the ROIs. This is conspicuous in the femoral neck, which has very few trabecular bones.

Moreover, the localized method also well depicts the characteristic trabecular patterns in the proximal femur with the conventional method as shown in Figure 5. The principal compressive group in the femoral head is well aligned along the hip contact force. The non-orthogonal intersection among the main trabeculae is well represented in the femoral neck. The orthogonal intersection among the secondary compressive and tensile groups is clearly described in intertrochanter. These results obviously indicate that the localized method has the same underlying principle as the conventional method. Note that topology optimization reconstructs bone microstructure to achieve maximum strength preserving minimum pixel-wise density difference based on the self-optimizing capabilities of bone.

The morphometric indices of the microstructure reconstructed using the localized method are almost consistent with those reconstructed using the conventional method, as shown in Table 2. The deviations of BV/TV are minimal, less than 0.28% for all ROIs. Interestingly, it was found that the ROI with higher BV/TV has smaller BV/TV errors for all ROIs. For example, the femoral neck with BV/TV of 20.9% showed the biggest error of 0.28% in the 4.8 × 4.8 mm^2^. Whereas the femoral head with BV/TV of 56.9% showed the lowest error of 0.02%. The Tb.Th, Tb.Sp, and Tb.N for the femoral head and intertrochanter showed a difference smaller than 6.9%. However, the femoral neck had larger index errors than the other ROIs. The Tb.Th, Tb.Sp, and Tb.N for the femoral neck region showed a maximum difference of 17.8%. This error increases as the size of the femoral neck decreases, although the absolute error is not large.

## 4. Discussion

Image resolution enhancement is a very important technique in the clinical field because it can provide more determinant information about bone health by enhancing a low-resolution (LR) image into a high-resolution (HR) image. One of the techniques, localized bone microstructure reconstruction based on topology optimization is the current only method that can reconstruct a bone microstructure HR image from its clinical routine LR image with high computational efficiency. However, this approach further needed thorough validation of the efficiency and accuracy of bone microstructure reconstruction compared with the conventional approach. As the essential step toward clinical application, this study validated the performance of the localized method by comparing it with the conventional method in terms of accuracy and efficiency of bone microstructure reconstruction. As a result, the localized method reconstructed nearly the same bone microstructure as the conventional method in a much shorter time.

This is the first study to investigate the computational efficiency of the localized bone microstructure reconstruction method by juxtaposing the results of the conventional and localized methods. The conventional method required at least 13.2 h and approximately 56.2 GB of memory for two-dimensional bone microstructure reconstruction. If reconstruction is extended to three dimensions, the total progress time and computational resource requirements increase, making its application to the clinical field, despite its high accuracy, difficult. Moreover, high-end computers have become indispensable owing to the massive computational resources required. However, this study demonstrated that the localized method can predict the bone microstructure in a shorter time, allowing the simultaneous analysis of bone microstructures in multiple ROIs. This possibility is due to at least 88.2% improvement in the total progress time and at least 95.9% improvement in the computational resources required. These computational efficiency results clearly demonstrate the clinical applicability of the topology optimization-based bone microstructure reconstruction method as a means of bone microstructure prediction.

This study quantitatively demonstrated that the localized method accurately expressed the bone microstructure with high accuracy and has a high predictive performance for the actual bone microstructure. The bone microstructure formed via the localized method showed a very similar structure to the conventional method with the main trabecular alignment angle alignment difference of less than 2.1° and an error rate of less than 17.8% in the morphometric indices. Although slight differences in the shape and angle of some side branches exist, the effect of these differences is negligible since the structure of the main trabecular bone along the main trajectory in each ROI is responsible for the structural behavior of the bone. The overall prediction performance in the femoral neck region was lower than that in the other regions. Note that the addition or deletion of a single strut in the area can be exaggerated since the femoral neck comprises of only a few thin bone microstructures [28]. However, as seen from the comparison by ROI size, the bone microstructure can be predicted sufficiently well if the size of the selected ROI is sufficiently large.

This study strengthened the reliability of the localized bone microstructure reconstruction method by following the first step of the systematic validation process. It is well known that the validation of bone metabolism-related simulation is conducted at four levels: proof-of-concept, case, population, and specimen-specific validation studies [27]. The first level, the prerequisite of all validation, investigates whether the proposed model produces similar patterns of actual bone distribution at large. As a proof-of-concept study of topology optimization-based localized bone microstructure reconstruction, this paper quantitively analyzed its reconstruction performance by comparing the conventional method. This study showed the results reconstructed by the localized method were similar in terms of the trabecular pattern and morphometric indices of the conventional method. Note that the conventional method has already been validated in a previous study [28] for its high predictive performance on the actual bone microstructure. The reconstructed trabecular bone had the characteristic trabecular patterns and the morphometric indices were in good agreement with the anatomical data in the literature. This implies that the localized method accurately reconstructs the actual trabecular bone microstructure with high resolution as well as observes the principle of the conventional method.

As mentioned previously, the bone microstructure results of the localized method showed good agreement with those of the conventional method. However, the microstructures reconstructed by the localized method were slightly different near the cut boundaries of the ROIs. This difference might be caused by the different resolutions of the load applied to the ROI in the two methods. In the conventional method, external global loads were applied to the cortical bone of the refined global model. Subsequently, the effect of loading is transmitted to the ROI. Thus, the resolution of the load transmitted to the ROI boundary is the same as that of the refined sub-element. In contrast, the localized method uses the estimated loads from the LR global model. This creates a different microstructure at the cut boundary compared with the conventional method since a load is applied for each size of the LR element. This phenomenon might be a numerical error due to Saint Venant’s principle [44]. This also affected the number of iterations; the localized method required more iterations than the conventional method.

The limitations of this study must be addressed. First, it validated the localized method in only two dimensions. For clinical applications, this validation should be expanded to three dimensions. Nevertheless, the results of this study show that computational efficiency can be improved sufficiently using a localized model. Second, this study used synthetic proximal femurs. The femur contains a bone microstructure that is generated via purely mechanical stimulus. However, the actual bone is modeled using both nonmechanical and mechanical stimuli. Bone-based validation is essential for exactness. Finally, this study used optimization parameters such as the sensitivity filtering radius and objective scaling factor as default values used in the literature [45,46]. However, these parameters were responsible for the efficiency of the optimization process. Thus, future work should consider optimization parameters for efficient bone microstructure reconstruction.

## 5. Conclusions

This study is the first study to quantitatively validate the topology optimization-based localized bone microstructure reconstruction method using the estimated local bone load for the ROI by comparing the computational efficiency and reconstruction accuracy. The morphometric indices, trabecular alignment angle, and trabecular bone shape and morphology of the localized method were in agreement with the conventional method. In addition, the localized method showed an improvement in computational efficiency of up to 88.2% compared to the conventional method. This implies that the localized method can simulate bone reconstruction with high computational efficiency and accuracy for the ROI. Additionally, it demonstrated that the localized method could predict the bone microstructure in a shorter time, allowing the simultaneous analysis of bone microstructures in multiple ROIs. Thus, the localized bone microstructure reconstruction method would contribute to improving the evaluation accuracy of bone strength in patients via bone microstructure prediction. In addition, this study could be an important foundation for further studies in the field of clinical radiology as well as biomedical engineering.

## Figures and Tables

**Figure 1 bioengineering-09-00644-f001:**
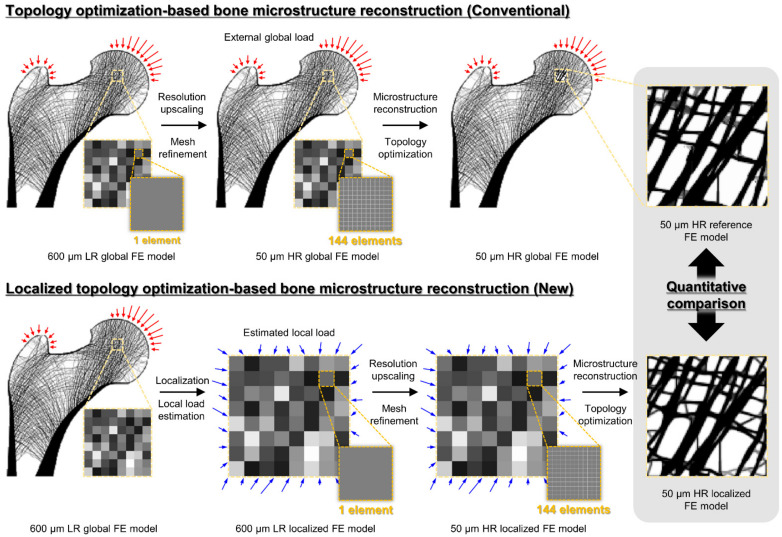
Overall validation procedure for the localized bone microstructure reconstruction method. The red arrows represent the external global load, and the blue arrows represent the estimated local load.

**Figure 2 bioengineering-09-00644-f002:**
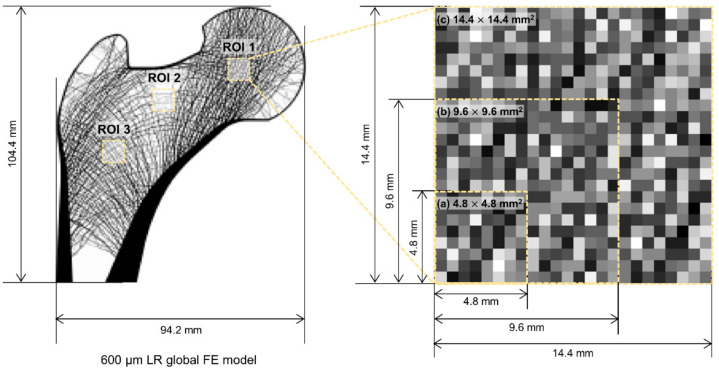
Low-resolution input model of a synthetic proximal femur with three regions of interest: femoral head (ROI 1), femoral neck (ROI 2), and intertrochanteric region (ROI 3).

**Figure 3 bioengineering-09-00644-f003:**
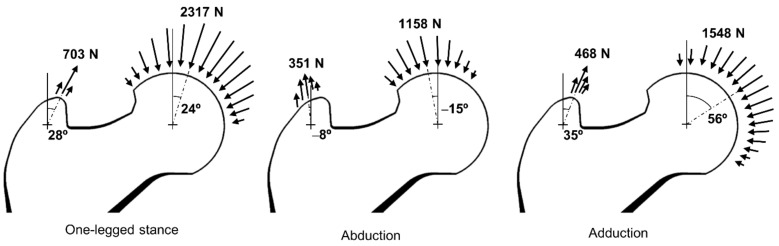
Three load conditions to consider daily activities.

**Figure 4 bioengineering-09-00644-f004:**
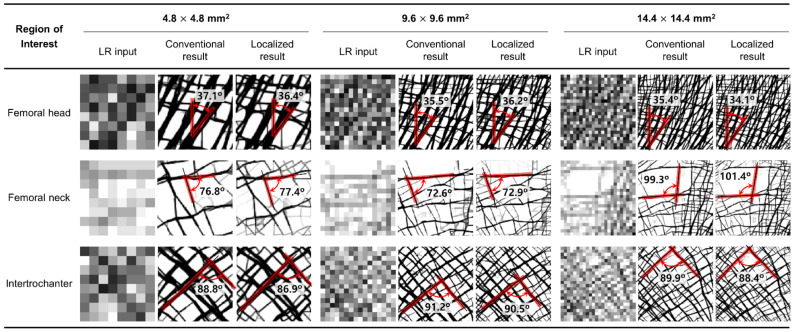
Trabecular architecture comparison of the low-resolution input (first column), the result of the conventional method (second column), and the result of the localized method (third column) according to the three different ROI locations and sizes.

**Figure 5 bioengineering-09-00644-f005:**
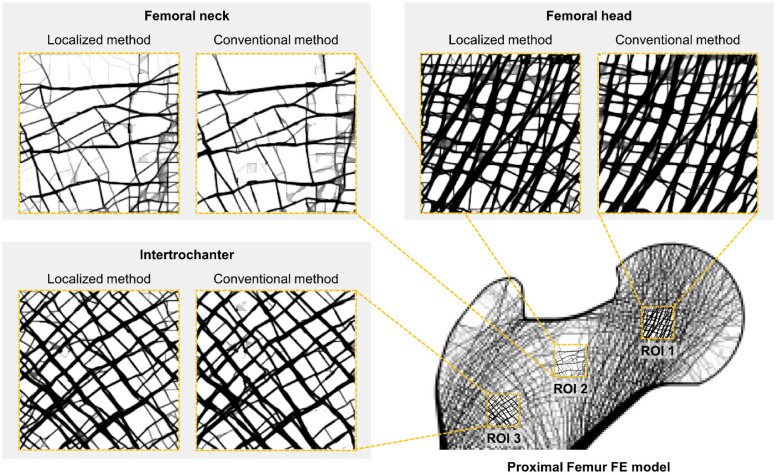
Bone microstructure reconstruction result of 14.4 × 14.4 mm^2^ by the conventional and localized method for the proximal femur.

**Table 1 bioengineering-09-00644-t001:** Comparison of the total progress time and required computational resources between the conventional and localized methods.

Region of Interest	Index	4.8 × 4.8 mm^2^			9.6 × 9.6 mm^2^			14.4 × 14.4 mm^2^		
		Conv. ^1^ Result	Loc. ^2^ Result	Imp. ^3^ (%)	Conv. ^1^ Result	Loc. ^2^ Result	Imp. ^3^ (%)	Conv. ^1^ Result	Loc. ^2^ Result	Imp. ^3^ (%)
Femoral head	Time (h)	22.8	2.0	91.23	21.6	1.3	93.98	22.8	2.7	88.16
	Iteration	363	689	-	320	457	-	351	607	-
	Resource (GB)	56.4	0.3	99.47	56.1	1.0	98.22	56.3	2.3	95.91
Femoral neck	Time (h)	13.3	0.7	94.74	13.2	0.8	93.94	21.7	1.8	91.71
	Iteration	200	233	-	210	289	-	278	474	-
	Resource (GB)	56.4	0.3	99.47	56.1	1.0	98.22	56.3	2.3	95.91
Intertrochanter	Time (h)	22.9	1.8	92.14	21.1	1.1	94.79	27.6	1.4	94.93
	Iteration	317	648	-	343	362	-	372	353	-
	Resource (GB)	56.4	0.3	99.47	56.1	1.0	98.22	56.3	2.3	95.91

^1^ Conv.: Conventional; ^2^ Loc.: Localized; ^3^ Imp.: Improvement.

**Table 2 bioengineering-09-00644-t002:** Comparison of trabecular morphometric indices between the conventional and localized method.

Region of Interest	Index	4.8 × 4.8 mm^2^			9.6 × 9.6 mm^2^			14.4 × 14.4 mm^2^		
		Conv. ^1^ Result	Loc. ^2^ Result	Error (%)	Conv. ^1^ Result	Loc. ^2^ Result	Error (%)	Conv. ^1^ Result	Loc. ^2^ Result	Error (%)
Femoral head	BV/TV (%)	56.89	56.87	0.02	56.41	56.37	0.04	54.96	54.93	0.03
	Tb.Th (μm)	262.65	245.34	6.59	268.84	255.73	4.88	249.06	241.76	2.93
	Tb.Sp (μm)	312.65	292.30	6.51	326.25	310.87	4.71	320.63	311.54	2.84
	Tb.N (mm^−1^)	2.17	2.32	6.91	2.10	2.20	4.76	2.21	2.27	2.71
Femoral neck	BV/TV (%)	21.14	20.86	0.28	20.02	19.93	0.09	22.30	22.21	0.09
	Tb.Th (μm)	113.97	95.81	15.93	125.18	110.21	11.96	141.04	132.16	6.30
	Tb.Sp (μm)	667.81	571.04	14.49	785.63	695.62	11.46	772.05	727.30	5.80
	Tb.N (mm^−1^)	1.85	2.18	17.84	1.60	1.81	13.12	1.58	1.68	6.33
Intertrochanter	BV/TV (%)	50.31	50.25	0.06	40.59	40.53	0.06	35.40	35.36	0.04
	Tb.Th (μm)	192.65	181.59	5.74	169.00	158.39	6.28	165.57	155.71	5.96
	Tb.Sp (μm)	298.85	282.43	5.49	388.62	365.00	6.08	474.70	447.04	5.83
	Tb.N (mm^−1^)	2.61	2.77	6.13	2.40	2.56	6.67	2.14	2.27	6.07

^1^ Conv.: Conventional; ^2^ Loc.: Localized.

## Data Availability

No new data were created or analyzed in this study. Data sharing is not applicable to this article.

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
