# Peer review of "Topology Optimization-Based Localized Bone Microstructure Reconstruction for Image Resolution Enhancement: Accuracy and Efficiency"

_bioengineering, 2022, doi:10.3390/bioengineering9110644_

Round 1
Reviewer 1 Report
In this article, topology optimization-based localized bone microstructure is validated by estimated local bone loads for the region of interest (ROI). The reconstructed results were compared for computational efficiency and reconstruction accuracy. Although the title of the manuscript is appropriate, these results and analysis are very superficial and short and are not enough to publish in the journal.
What is the main novelty of the research? The novelty should be presented in the abstract and conclusion. What are the advantages of using the localized reconstruction method? If it is not accurate enough, why is it used? How are the results validated?
The abstract has almost no quantitative data. It is better to provide quantitative data from the comparison of methods.
The number of keywords used is very high. Some of them are not mentioned at all in the abstract.
The results and conclusions are similar. Both sections provide the same limited results.
The discussion and conclusion section should be separate. Of course, this section had no discussion and analysis of results.
Analysis of the results should be added to the results section or presented in a separate section.
Reviewer 2 Report
1. please avoid using we in the text, such as in lines 12, and 15.
2. Separate conclusion from the discussion.
3. Stability of the results should be ensured by using the convergence study for the Finite element meshes.
4. Loading conditions and their values on the stance, abduction and adduction conditions should be clearly stated.
Reviewer 3 Report
This study aims to validate the reconstruction method using the estimated local bone loads for the region of interest (ROI). In this aspect, introduction of this paper provides sufficient background and include all relevant references. Also, I think that all the cited references of 45ea relevant to the research.
In the section 2, materials & methods include conventional topology optimization-based bone microstructure reconstruction using the global model, novel topology optimization-based localized bone microstructure reconstruction using the localized model, numerical validation based on quantitative comparison using proximal femur. This proves that the study design is appropriate and the methods adequately described.
In the section 3, 4, Upon comparing the morphometric indices and trabecular alignment angle, this result observed that the trabecular bone shape and morphology obtained using the localized method were very similar to those obtained using the conventional method. In addition, the localized method results showed an 88% improvement in computational efficiency compared to the conventional method. I think that the results clearly are presented and the conclusions supported by the results.
Based on the results of this study, I expect that localized bone microstructure reconstruction methods will improve the evaluation accuracy of bone strength in patients via bone microstructure prediction.
Reviewer 4 Report
please delete the first 2 sentences..."A recent study proposed a topology optimization-based localized bone microstructure 8 reconstruction method to reduce the excessive computational resources and time required due to 9 optimization. However, it did not deal with the efficiency and accuracy of that approach compared 10 with the conventional approach.".. these sentences seem problem statements and it is not a good practice to refer to some recent paper in the abstract
- please divide the conclusions into two sections, i.e., 1) discussions and analysis 2) conclusion
- overall paper is in good shape
Round 2
Reviewer 1 Report
The quality of the revised manuscript has improved, but the novelty of the work, the analysis of the results, and the need for more data are still criticized.
Some comments have not been fully considered. For example, image upscaling is mentioned once in the entire article, and it is also a keyword.
Author Response
"Please see the attachment.
